# A Chain Graph Interpretation
# of Real-World Neural Networks

## Abstract

The last decade has witnessed a boom of deep learning research and applications achieving state-of-the-art results in various domains. However, most advances have been established empirically, and their theoretical analysis remains lacking. One major issue is that our current interpretation of neural networks (NNs) as function approximators is too generic to support in-depth analysis. In this paper, we remedy this by proposing an alternative interpretation that identifies NNs as chain graphs (CGs) and feed-forward as an approximate inference procedure. The CG interpretation specifies the nature of each NN component within the rich theoretical framework of probabilistic graphical models, while at the same time remains general enough to cover real-world NNs with arbitrary depth, multi-branching and varied activations, as well as common structures including convolution / recurrent layers, residual block and dropout. We demonstrate with concrete examples that the CG interpretation can provide novel theoretical support and insights for various NN techniques, as well as derive new deep learning approaches such as the concept of partially collapsed feed-forward inference. It is thus a promising framework that deepens our understanding of neural networks and provides a coherent theoretical formulation for future deep learning research.

## 1 Introduction

During the last decade, deep learning (Goodfellow et al., 2016), the study of neural networks (NNs), has achieved ground-breaking results in diverse areas such as computer vision (Krizhevsky et al., 2012; He et al., 2016; Long et al., 2015; Chen et al., 2018), natural language processing (Hinton et al., 2012; Vaswani et al., 2017; Devlin et al., 2019), generative modeling (Kingma & Welling, 2014; Goodfellow et al., 2014) and reinforcement learning (Mnih et al., 2015; Silver et al., 2016), and various network designs have been proposed. However, neural networks have been treated largely as "black-box" function approximators, and their designs have chiefly been found via trial-and-error, with little or no theoretical justification. A major cause that hinders the theoretical analysis is the current overly generic modeling of neural networks as function approximators: simply interpreting a neural network as a composition of parametrized functions provides little insight to decipher the nature of its components or its behavior during the learning process.

In this paper, we show that a neural network can actually be interpreted as a probabilistic graphical model (PGM) called chain graph (CG) (Koller & Friedman, 2009), and feed-forward as an efficient approximate probabilistic inference on it. This offers specific interpretations for various neural network components, allowing for in-depth theoretical analysis and derivation of new approaches.

### 1.1 Related work

In terms of theoretical understanding of neural networks, a well known result based on the function approximator view is the universal approximation theorem (Goodfellow et al., 2016), however it only establishes the representational power of NNs. Also, there have been many efforts on alternative NN interpretations. One prominent approach identifies infinite width NNs as Gaussian processes (Neal, 1996; Lee et al., 2018), enabling kernel method analysis (Jacot et al., 2018). Other works also employ theories such as optimal transport (Genevay et al., 2017; Chizat & Bach, 2018) or mean field (Mei et al., 2019). These approaches lead to interesting findings, however they tend to only hold under limited or unrealistic settings and have difficulties interpreting practical real-world NNs.

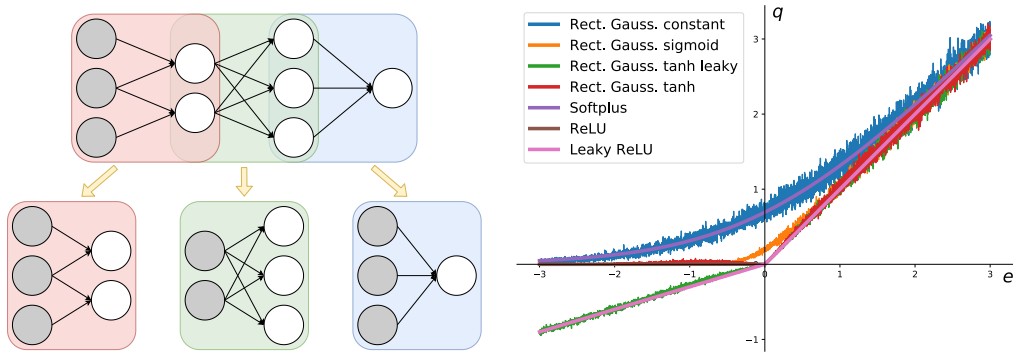

Figure 1: Neural networks can be interpreted as layered chain graphs where activation functions are determined by node distributions. *Left*: An example neural network interpreted as a chain graph with three chain components which represent its layers; *Right*: A variety of activation functions (softplus, ReLU, leaky ReLU) approximated by nodes following rectified Gaussian distributions ($e, q$ as in Eq. (7)). We visualize the approximations stochastically by averaging over 200 samples.

Alternatively, some existing works study the post-hoc interpretability (Lipton, 2018), proposing methods to analyze the empirical behavior of trained neural networks: activation maximization (Erhan et al., 2009), typical input synthesis (Nguyen et al., 2016), deconvolution (Zeiler & Fergus, 2014), layer-wise relevance propagation (Bach et al., 2015), etc. These methods can offer valuable insights to the practical behavior of neural networks, however they represent distinct approaches and focuses, and are all limited within the function approximator view.

Our work links neural networks to probabilistic graphical models (Koller & Friedman, 2009), a rich theoretical framework that models and visualizes probabilistic systems composed of random variables (RVs) and their interdependencies. There are several types of graphical models. The chain graph model (also referred to as the LWF chain graph model) (Koller & Friedman, 2009; Lauritzen & Wermuth, 1989; Frydenberg, 1990) used in our work is a general form that unites directed and undirected variants, visualized as a partially directed acyclic graph (PDAG). Interestingly, there exists a series of works on constructing hierarchical graphical models for data-driven learning problems, such as sigmoid belief network (Neal, 1992), deep belief network (Hinton et al., 2006), deep Boltzmann machine (Salakhutdinov & Hinton, 2012) and sum product network (Poon & Domingos, 2011). As alternatives to neural networks, these models have shown promising potentials for generative modeling and unsupervised learning. Nevertheless, they are yet to demonstrate competitive performances over neural network for discriminative learning.

Neural networks and graphical models have so far been treated as two distinct approaches in general. Existing works that combine them (Zheng et al., 2015; Chen et al., 2018; Lample et al., 2016) mainly treat either neural networks as function approximators for amortized inference, or graphical models as post-processing steps. Tang & Salakhutdinov (2013) create a hybrid model, the stochastic feedforward neural network (SFNN), by concatenating deterministic neurons with stochastic Bernoulli random variables, in order to represent multimodal distributions. Some also consider neural networks as graphical models with deterministic hidden nodes (Buntine, 1994). However this is an atypical degenerate regime. To the best of our knowledge, our work provides the first rigorous and comprehensive formulation of a (non-degenerate) graphical model interpretation for neural networks in practical use.

## 1.2 OUR CONTRIBUTIONS

The main contributions of our work are summarized as follows:

- We propose a layered chain graph representation of neural networks, interpret feed-forward as an approximate probabilistic inference procedure, and show that this interpretation provides an extensive coverage of practical NN components (Section 2);

- To illustrate its advantages, we show with concrete examples (residual block, RNN, dropout) that the chain graph interpretation enables coherent and in-depth theoretical support, and provides additional insights to various empirically established network structures (Section 3);
- Furthermore, we demonstrate the potential of the chain graph interpretation for discovering new approaches by using it to derive a novel stochastic inference method named partially collapsed feed-forward, and establish experimentally its empirical effectiveness (Section 4).

## 2 CHAIN GRAPH INTERPRETATION OF NEURAL NETWORKS

Without further delay, we derive the chain graph interpretation of neural networks in this section. We will state and discuss the main results here and leave the proofs in the appendix.

### 2.1 THE LAYERED CHAIN GRAPH REPRESENTATION

We start by formulating the so called *layered chain graph* that corresponds to neural networks we use in practice: Consider a system represented by $L$ layers of random variables $(\mathbf{X}^1, \ldots, \mathbf{X}^L)$, where $X_i^l$ is the $i$-th variable node in the $l$-th layer, and denote $N^l$ the number of nodes in layer $l$. We assume that nodes $X_i^l$ in the same layer $l$ have the same distribution type characterized by a feature function $\mathbf{T}^l$ that can be multidimensional. Also, we assume that the layers are ordered topologically and denote $Pa(\mathbf{X}^l)$ the parent layers of $\mathbf{X}^l$. To ease our discussion, we assume that $\mathbf{X}^1$ is the input layer and $\mathbf{X}^L$ the output layer (our formulation can easily extend to multi-input/output cases). A layered chain graph is then defined as follows:

**Definition 1.** A *layered chain graph* that involves $L$ layers of random variables $(\mathbf{X}^1, \ldots, \mathbf{X}^L)$ is a chain graph that encodes the overall distribution $P(\mathbf{X}^2, \ldots, \mathbf{X}^L | \mathbf{X}^1)$ such that:

1. It can be factored into layerwise chain components $P(\mathbf{X}^l | Pa(\mathbf{X}^l))$ following the topological order, and nodes $X_i^l$ within each chain component $P(\mathbf{X}^l | Pa(\mathbf{X}^l))$ are conditionally independent given their parents (this results in bipartite chain components), thus allowing for further decomposition into nodewise conditional distributions $P(X_i^l | Pa(\mathbf{X}^l))$ . This means we have

$$P(\mathbf{X}^2, \ldots, \mathbf{X}^L | \mathbf{X}^1) = \prod_{l=2}^{L} P(\mathbf{X}^l | Pa(\mathbf{X}^l)) = \prod_{l=2}^{L} \prod_{i=1}^{N^l} P(X_i^l | Pa(\mathbf{X}^l)); \tag{1}$$

2. For each layer $l$ with parent layers $Pa(\mathbf{X}^l) = \{\mathbf{X}^{p_1}, \ldots \mathbf{X}^{p_n}\}, p_1, \ldots, p_n \in \{1, \ldots, l-1\}$, its nodewise conditional distributions $P(X_i^l | Pa(\mathbf{X}^l))$ are modeled by pairwise conditional random fields (CRFs) with with unary ($\mathbf{b}_i^l$) and pairwise ($\mathbf{W}_{j,i}^{p,l}$) weights (as we will see, they actually correspond to biases and weights in NN layers):

$$P(X_i^l | Pa(\mathbf{X}^l)) = f^l\big(\mathbf{T}^l(X_i^l), \mathbf{e}_i^l\big(\mathbf{T}^{p_1}(\mathbf{X}^{p_1}), \ldots, \mathbf{T}^{p_n}(\mathbf{X}^{p_n})\big)\big) \tag{2}$$

$$\text{with} \quad \mathbf{e}_i^l\big(\mathbf{T}^{p_1}(\mathbf{X}^{p_1}), \ldots, \mathbf{T}^{p_n}(\mathbf{X}^{p_n})\big) = \mathbf{b}_i^l + \sum_{p=p_1}^{p_n} \sum_{j=1}^{N^p} \mathbf{W}_{j,i}^{p,l} \mathbf{T}^p(X_j^p). \tag{3}$$

Figure 1 Left illustrates an example three-layer network as layered chain graph and its chain component factorization. In Eq. (2), $f^l$ is an arbitrary function that represents a probability distribution. For exponential family distributions (Koller & Friedman, 2009), Eq. (2) simply becomes $P(X_i^l | Pa(\mathbf{X}^l)) \propto \exp\big(\mathbf{T}^l(X_i^l) \cdot \mathbf{e}_i^l\big(\mathbf{T}^{p_1}(\mathbf{X}^{p_1}), \ldots, \mathbf{T}^{p_n}(\mathbf{X}^{p_n})\big)\big)$.

Note that layered chain graph has a globally directed graph structure and has an equivalent modeling based on directed graphical model (Bayesian network) (Koller & Friedman, 2009), we elaborate on this point for interested readers in Appendix A.

### 2.2 FEED-FORWARD AS APPROXIMATE PROBABILISTIC INFERENCE

To identify layered chain graphs with real-world neural networks, we need to show that they can behave the same way during inference and learning. For this, we establish the fact that feed-forward can actually be seen as performing an approximate probabilistic inference on a layered chain graph:

Given an input sample $\tilde{\mathbf{x}}^1$, we consider the problem of inferring the marginal distribution $Q_i^l$ of a node $X_i^l$ and its expected features $\mathbf{q}_i^l$, defined as

$$Q_i^l(x_i^l|\tilde{\mathbf{x}}^1) = P(X_i^l = x_i^l|\mathbf{X}^1 = \tilde{\mathbf{x}}^1); \quad \mathbf{q}_i^l = \mathbb{E}_{Q_i^l}[\mathbf{T}^l(X_i^l)] \ (\mathbf{q}^1 = \tilde{\mathbf{x}}^1). \tag{4}$$

Consider a non-input layer $l$ with parent layers $p_1, \ldots, p_n$, the independence assumptions encoded by the layered chain graph lead to the following recursive expression for marginal distributions $Q$:

$$Q_i^l(x_i^l|\tilde{\mathbf{x}}^1) = \mathbb{E}_{Q^{p_1}, \ldots, Q^{p_n}}[P(x_i^l|Pa(\mathbf{X}^l))]. \tag{5}$$

However, the above expression is in general intractable, as it integrates over the entire admissible states of all parents nodes in $Pa(\mathbf{X}^l)$. To proceed further, simplifying approximations are needed. Interestingly, by using linear approximations, we can obtain the following results (in case of discrete random variable the integration in Eq. 7 is replaced by summation):

**Proposition 1.** *If we make the assumptions that the corresponding expressions are approximately linear w.r.t. parent features $\mathbf{T}^{p_1}(\mathbf{X}^{p_1}), \ldots, \mathbf{T}^{p_n}(\mathbf{X}^{p_n})$, we obtain the following approximations:*

$$Q_i^l(x_i^l|\tilde{\mathbf{x}}^1) \approx f^l(\mathbf{T}^l(x_i^l), \mathbf{e}_i^l(\mathbf{q}^{p_1}, \ldots, \mathbf{q}^{p_n})); \tag{6}$$

$$\mathbf{q}_i^l \approx \int_{x_i^l} \mathbf{T}^l(x_i^l) f^l(\mathbf{T}^l(x_i^l), \mathbf{e}_i^l(\mathbf{q}^{p_1}, \ldots, \mathbf{q}^{p_n})) dx_i^l := \mathbf{g}^l(\mathbf{e}_i^l(\mathbf{q}^{p_1}, \ldots, \mathbf{q}^{p_n})). \tag{7}$$

*Especially, Eq. (7) is a feed-forward expression for expected features $\mathbf{q}_i^l$ with activation function $\mathbf{g}^l$ determined by $\mathbf{T}^l$ and $f^l$, i.e. the distribution type of random variable nodes in layer $l$.*

The proof is provided in Appendix B.1. This allows us to identify feed-forward as an approximate probabilistic inference procedure for layered chain graphs. For learning, the loss function is typically a function of $(Q^L, \mathbf{q}^L)$ obtainable via feed-forward, and we can follow the same classical neural network parameter update using stochastic gradient descent and backpropagation. Thus we are able to replicate the exact neural network training process with this layered chain graph framework.

The following corollary provides concrete examples of some common activation functions $\mathbf{g}$ (we emphasize their names in bold, detailed formulations and proofs are given in Appendix B.2):

**Corollary 2.** *We have the following node distribution - activation function correspondences:*

1. *Binary nodes taking values $\{\alpha, \beta\}$ results in sigmoidal activations, especially, we obtain **sigmoid** with $\alpha = 0, \beta = 1$ and **tanh** with $\alpha = -1, \beta = 1$ ($\alpha, \beta$ are interchangeable);*
2. *Multilabel nodes characterized by label indicator features result in the **softmax** activation;*
3. *Variants of (leaky) rectified Gaussian distributions ($T_i^l(X_i^l) = X_i^l = \max(\epsilon Y_i^l, Y_i^l)$ with $Y_i^l \sim \mathcal{N}(e_i^l, (s_i^l(e_i^l))^2)$) can approximate activations such as **softplus** ($\epsilon = 0, s_i^l \approx 1.7761$) and $\epsilon$-**leaky rectified linear unit** (ReLU) ($s_i^l = \tanh(e_i^l)$) including **ReLU** ($\epsilon = 0$) and **identity** ($\epsilon = 1$).*

Figure 1 Right illustrates activation functions approximated by various rectified Gaussian variants. We also plotted (in orange) an alternative approximation of ReLU with sigmoid-modulated standard deviation proposed by Nair & Hinton (2010) which is less accurate around the kink at the origin.

The linear approximations, needed for feed-forward, is coarse and only accurate for small pairwise weights ($\|\mathbf{W}\| \ll 1$) or already linear regions. This might justify weight decay beyond the general "anti-overfit" argument and the empirical superiority of piecewise linear activations like ReLU (Nair & Hinton, 2010). Conversely, as a source of error, it might explain some "failure cases" of neural networks such as their vulnerability against adversarial samples, see e.g., Goodfellow et al. (2015).

### 2.3 GENERALITY OF THE CHAIN GRAPH INTERPRETATION

The chain graph interpretation formulated in Sections 2.1 and 2.2 is a general framework that can describe many practical network structures. To demonstrate this, we list here a wide range of neural network designs (marked in bold) that are chain graph interpretable.

- In terms of network architecture, it is clear that the chain graph interpretation can model networks of arbitrary depth, and with general multi-branched structures such as **inception modules** (Szegedy et al., 2015) or **residual blocks** (He et al., 2016; He et al., 2016) discussed in Section 3.1. Also, it is possible to built up **recurrent neural networks (RNNs)** for sequential data

learning, as we will see in Section 3.2. Furthermore, the modularity of chain components justifies **transfer learning via partial reuse of pre-trained networks**, e.g., backbones trained for image classification can be reused for segmentation (Chen et al., 2018).

- In terms of layer structure, we are free to employ sparse connection patterns and shared/fixed weight, so that we can obtain not only **dense connections**, but also connections like **convolution**, **average pooling** or **skip connections**. Moreover, as shown in Section 3.3, **dropout** can be reproduced by introducing and sampling from auxiliary random variables, and normalization layers like **batch normalization** (Ioffe & Szegedy, 2015) can be seen as reparametrizations of node distributions and fall within the general form (Eq. (2)). Finally, we can extend the layered chain graph model to allow for intra-layer connections, which enables **non-bipartite CRF layers** which are typically used on output layers for structured prediction tasks like image segmentation (Zheng et al., 2015; Chen et al., 2018) or named entity recognition (Lample et al., 2016). However, feed-forward is no longer applicable through these intra-connected layers.
- Node distributions can be chosen freely, leading to a variety of nonlinearities (e.g., Corollary 2).

## 3 SELECTED CASE STUDIES OF EXISTING NEURAL NETWORK DESIGNS

The proposed chain graph interpretation offers a detailed description of the underlying mechanism of neural networks. This allows us to obtain novel theoretical support and insights for various network designs which are consistent within a unified framework. We illustrate this with the following concrete examples where we perform in-depth analysis based on the chain graph formulation.

### 3.1 RESIDUAL BLOCK AS REFINEMENT MODULE

The residual block, proposed originally in He et al. (2016) and improved later (He et al., 2016) with the preactivation form, is an effective design for building up very deep networks. Here we show that a preactivation residual block corresponds to a refinement module within a chain graph. We use *modules* to refer to encapsulations of layered chain subgraphs as input–output mappings without specifying their internal structures. A refinement module is defined as follows:

**Definition 2.** Given a base submodule from layer $\mathbf{X}^{l-1}$ to layer $\mathbf{X}^l$, a *refinement module* augments this base submodule with a side branch that chains a copy of the base submodule (sharing weight with its original) from $\mathbf{X}^{l-1}$ to a duplicated layer $\tilde{\mathbf{X}}^l$, and then a refining submodule from $\tilde{\mathbf{X}}^l$ to $\mathbf{X}^l$.

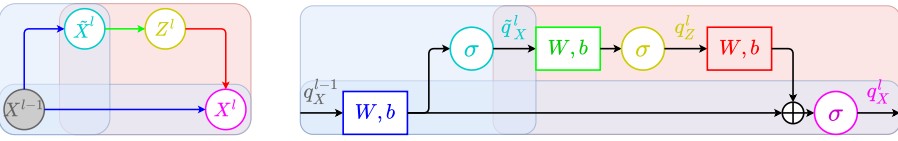

Figure 2: Example of a refinement module (left) and its corresponding computational graph (right), composed of a base submodule $X^{l-1} \to X^l$ (blue background) and a refining submodule $\tilde{X}^l \to Z^l \to X^l$ (red background). In the computational graph each $W, b$ represents a linear connection (Eq. (3)) and $\sigma$ an activation function. Same color identifies corresponding parts in the two graphs. We see that this refinement module corresponds exactly to a preactivation residual block.

**Proposition 3.** *A refinement module corresponds to a preactivation residual block.*

We provide a proof in Appendix B.3 and illustrate this correspondence in Figure 2. An interesting remark is that the refinement process can be recursive: the base submodule of a refinement module can be a refinement module itself. This results in a sequence of consecutive residual blocks.

While a vanilla layered chain component encodes a generalized linear model during feed-forward (c.f. Eqs. (7),(3)), the refinement process introduces a nonlinear extension term to the previously linear output preactivation, effectively increasing the representational power. This provides a possible explanation to the empirical improvement generally observed when using residual blocks.

Note that it is also possible to interpret the original postactivation residual blocks, however in a somewhat artificial manner, as it requires defining identity connections with manually fixed weights.

### 3.2 RECURRENT NEURAL NETWORKS

Recurrent neural networks (RNNs) (Goodfellow et al., 2016) are widely used for handling sequential data. An unrolled recurrent neural network can be interpreted as a dynamic layered chain graph constructed as follows: a given base layered chain graph is copied for each time step, then these copies are connected together through recurrent chain components following the Markov assumption (Koller & Friedman, 2009): each recurrent layer $\mathbf{X}^{l,t}$ at time $t$ is connected by its corresponding layer $\mathbf{X}^{l,t-1}$ from the previous time step $t-1$. Especially, denoting $Pa^t(\mathbf{X}^{l,t})$ the non-recurrent parent layers of $\mathbf{X}^{l,t}$ in the base chain graph, we can easily interpret the following two variants:

**Proposition 4.** *Given a recurrent chain component that encodes $P(\mathbf{X}^{l,t}|Pa^t(\mathbf{X}^{l,t}), \mathbf{X}^{l,t-1})$,*

1. *It corresponds to a simple (or vanilla / Elman) recurrent layer (Goodfellow et al., 2016) if the connection from $\mathbf{X}^{l,t-1}$ to $\mathbf{X}^{l,t}$ is dense;*
2. *It corresponds to an independently RNN (IndRNN) (Li et al., 2018) layer if the conditional independence assumptions among the nodes $X_i^{l,t}$ within layer $l$ are kept through time:*

$$\forall i \in \{1, \ldots, N^l\}, \ P(X_i^{l,t}|Pa^t(\mathbf{X}^{l,t}), \mathbf{X}^{l,t-1}) = P(X_i^{l,t}|Pa^t(\mathbf{X}^{l,t}), X_i^{l,t-1}). \tag{8}$$

We provide a proof in Appendix B.4 and illustrates both variants in Figure 3.

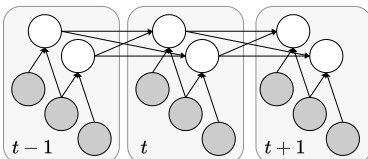 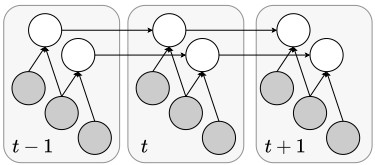

Figure 3: Comparison of an simple recurrent layer (left) v.s. an IndRNN (right) recurrent layer. IndRNN, the better variant, enforces the intra-layer conditional independence through time.

The simple recurrent layer, despite its exhaustive dense recurrent connection, is known to suffer from vanishing/exploding gradient and can not handle long sequences. The commonly used long-short term memory (Hochreiter & Schmidhuber, 1997) and gated recurrent unit (Cho et al., 2014) alleviate this issue via long term memory cells and gating. However, they tend to result in bloated structures, and still cannot handle very long sequences (Li et al., 2018). On the other hand, IndRNNs can process much longer sequences and significantly outperform not only simple RNNs, but also LSTM-based variants (Li et al., 2018; 2019). This indicates that the assumption of intra-layer conditional independence through time, analogue to the local receptive fields of convolutional neural networks, could be an essential sparse network design tailored for sequential modeling.

### 3.3 DROPOUT

Dropout (Srivastava et al., 2014) is a practical stochastic regularization method commonly used especially for regularizing fully connected layers. As we see in the following proposition, from the chain graph point of view, dropout corresponds to introducing Bernoulli auxiliary random variables that serve as noise generators for feed-forward during training:

**Proposition 5.** *Adding dropout with drop rate $1 - p^l$ to layer $l$ corresponds to the following chain graph construction: for each node $X_i^l$ in layer $l$ we introduce an auxiliary Bernoulli random variable $D_i^l \sim Bernoulli(p^l)$ and multiply it with the pairwise interaction terms in all preactivations (Eq. (3)) involving $X_i^l$ as parent (this makes $D_i^l$ a parent of all child nodes of $X_i^l$ and extend their pairwise interactions with $X_i^l$ to ternary ones). The behavior of dropout is reproduced exactly if:*

- *During training, we sample auxiliary nodes $D_i^l$ during each feed-forward. This results in dropping each activation $\mathbf{q}_i^l$ of node $X_i^l$ with probability $1 - p^l$;*
- *At test time, we marginalize auxiliary nodes $D_i^l$ during each feed-forward. This leads to deterministic evaluations with a constant scaling of $p^l$ for the node activations $\mathbf{q}_i^l$.*

We provide a proof in Appendix B.5. Note that among other things, this chain graph interpretation of dropout provides a theoretical justification of the constant scaling at test time. This was originally proposed as a heuristic in Srivastava et al. (2014) to maintain consistent behavior after training.

## 4 PARTIALLY COLLAPSED FEED-FORWARD

The theoretical formulation provided by the chain graph interpretation can also be used to derive new approaches for neural networks. It allows us to create new deep learning methods following a coherent framework that provides specific semantics to the building blocks of neural networks. Moreover, we can make use of the abundant existing work from the PGM field, which also serves as a rich source of inspiration. As a concrete example, we derive in this section a new stochastic inference procedure called partially collapsed feed-forward (PCFF) using the chain graph formulation.

### 4.1 PCFF: CHAIN GRAPH FORMULATION

A layered chain graph, which can represent a neural network, is itself a probabilistic graphical model that encodes an overall distribution conditioned on the input. This means that, to achieve stochastic behavior, we can directly draw samples from this distribution, instead of introducing additional "noise generators" like in dropout. In fact, given the globally directed structure of layered chain graph, and the fact that the conditioned input nodes are ancestral nodes without parent, it is a well-known PGM result that we can apply forward sampling (or ancestral sampling) (Koller & Friedman, 2009) to efficiently generate samples: given an input sample $\tilde{\mathbf{x}}^1$, we follow the topological order and sample each non-input node $X_i^l$ using its nodewise distribution (Eq. (2)) conditioned on the samples $(\mathbf{x}^{p_1}, \ldots, \mathbf{x}^{p_n})$ of its parents. Compared to feed-forward, forward sampling also performs a single forward pass, but generates instead an unbiased stochastic sample estimate.

While in general an unbiased estimate is preferable and the stochastic behavior can also introduce regularization during training (Srivastava et al., 2014), forward sampling can not directly replace feed-forward, since the sampling operation is not differentiable and will jeopardize the gradient flow during backpropagation. To tackle this, one idea is to apply the reparametrization trick (Kingma & Welling, 2014) on continuous random variables (for discrete RVs the Gumbel softmax trick (Jang et al., 2017) can be used but requires additional continuous relaxation). An alternative solution is to only sample part of the nodes as in the case of dropout.

The proposed partially collapse feed-forward follows the second idea: we simply "mix up" feed-forward and forward sampling, so that for each forward inference during training, we randomly select a portion of nodes to sample and the rest to compute deterministically with feed-forward. Thus for a node $X_i^l$ with parents $(\mathbf{X}^{p_1}, \ldots, \mathbf{X}^{p_n})$, its forward inference update becomes

$$\mathbf{q}_i^l \leftarrow \begin{cases} \mathbf{g}^l(\mathbf{e}_i^l(\mathbf{q}^{p_1}, \ldots, \mathbf{q}^{p_n})) & \text{if collapsed (feed-forward);} \\ \mathbf{T}^l(x_i^l), \ x_i^l \sim f^l\big(\mathbf{T}^l(X_i^l), \mathbf{e}_i^l(\mathbf{q}^{p_1}, \ldots, \mathbf{q}^{p_n})\big) & \text{if uncollapsed (forward sampling).} \end{cases} \quad (9)$$

Following the collapsed sampling (Koller & Friedman, 2009) terminology, we call this method the partially collapsed feed-forward (PCFF). PCFF is a generalization over feed-forward and forward sampling, which can be seen as its fully collapsed / uncollapsed extremes. Furthermore, it offers a bias–variance trade-off, and can be combined with the reparametrization trick to achieve unbiased estimates with full sampling, while simultaneously maintaining the gradient flow.

**Relation to stochastic feedforward neural network**    While PCFF can also be seen as a stochastic generalization of the feed-forward inference, it represents a substantially distinct approach compared to SFNN: Apart from the clear difference that PCFF uses forward sampling and SFNN uses importance sampling, a major dissimilarity is that SFNN makes a clear distinction between deterministic neurons and stochastic random variables, whereas PCFF identifies neurons with random variables thanks to the layered chain graph interpretation. This is why PCFF can freely choose a different subset of nodes to sample during each forward pass. From the chain graph interpretation perspective, SFNN can be seen as a layered chain graph having a fixed subset of nodes with stochastic behavior, and it performs a hybrid of feed-forward and importance sampling for inference.

### 4.2 PCFF: EXPERIMENTAL VALIDATION

In the previous sections, we have been discussing existing approaches whose empirical evaluations have been thoroughly covered by prior work. The novel PCFF approach proposed in this section, however, requires experiments to check its practical effectiveness. For this we conduct here a series

of experiments[1]. Our emphasis is to understand the behavior of PCFF under various contexts and not to achieve best result for any specific task. We only use chain graph interpretable components, and we adopt the reparameterization trick (Kingma & Welling, 2014) for ReLU PCFF samples.

The following experiments show that PCFF is overall an effective stochastic regularization method. Compared to dropout, it tends to produce more consistent performance improvement, and can sometimes outperform dropout. This confirms that our chain graph based reasoning has successfully found an interesting novel deep learning method.

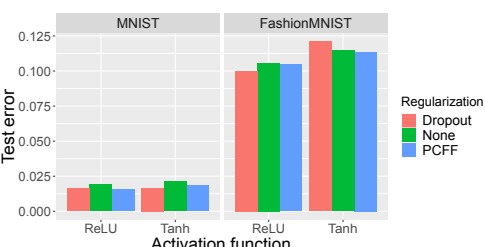 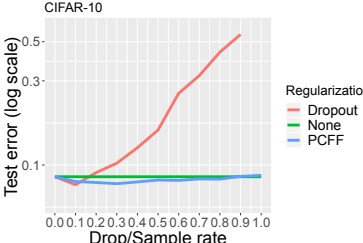

Figure 4: Comparison of stochastic methods (None/Dropout/PCFF) in terms of image classification test errors (lower is better) under various settings. Left: MNIST/FashionMNIST datasets with a simple dense network and tanh/ReLU activation functions; Right: CIFAR-10 dataset with ResNet20 and varying drop/sample rates. All reported results are average values of three runs. Compared to dropout, PCFF can achieve comparable results, and tend to deliver more consistent improvements.

**Simple dense network**  We start with a simple network with two dense hidden layers of 1024 nodes to classify MNIST (Lecun et al., 1998) and FashionMNIST (Xiao et al., 2017) images. We use PyTorch (Paszke et al., 2017), train with stochastic gradient descent (learning rate 0.01, momentum 0.9), and set up 20% of training data as validation set for performance monitoring and early-stopping. We set drop rate to 0.5 for dropout, and for PCFF we set the sample rate to 0.4 for tanh and 1.0 (full sampling) for ReLU. Figure 4 Left reports the test errors with different activation functions and stochastic regularizations.

We see that dropout and PCFF are overall comparable, and both improve the results in most cases. Also, the ReLU activation consistently produces better results that tanh. Additional experiments show that PCFF and dropout can be used together, which sometimes yields improved performance.

**Convolutional residual network**  To figure out the applicability of PCFF in convolutional residual networks, we experiment on CIFAR-10 (Krizhevsky, 2009) image classification. For this we adapt an existing implementation (Idelbayev) to use the preactivation variant. We focus on the ResNet20 structure, and follow the original learning rate schedule except for setting up a validation set of 10% training data to monitor training performance. Figure 4 Right summarizes the test errors under different drop/sample rates.

We observe that in this case PCFF can improve the performance over a wide range of sample rates, whereas dropout is only effective with drop rate 0.1, and large drop rates in this case significantly deteriorate the performance. We also observe a clear trade-off of the PCFF sample rate, where a partial sampling of 0.3 yields the best result.

**Independently RNN**  We complete our empirical evaluations of PCFF with an RNN test case. For this we used IndRNNs with 6 layers to solve the sequential/permuted MNIST classification problems based on an existing Implementation[2] provided by the authors of IndRNN (Li et al., 2018; 2019). We tested over dropout with drop rate 0.1 and PCFF with sample rate 0.1 and report the average test accuracy of three runs. We notice that, while in the permuted MNIST case both dropout (0.9203) and PCFF (0.9145) improves the result (0.9045), in the sequential MNIST case, dropout (0.9830) seems to worsen the performance (0.9841) whereas PCFF (0.9842) delivers comparable result.

---

[1]Implementation available at: (Github link placeholder, provided as supplementary material.)

[2]https://github.com/Sunnydreamrain/IndRNN_pytorch

## 5 CONCLUSIONS AND DISCUSSIONS

In this work, we show that neural networks can be interpreted as layered chain graphs, and that feed-forward can be viewed as an approximate inference procedure for these models. This chain graph interpretation provides a unified theoretical framework that elucidates the underlying mechanism of real-world neural networks and provides coherent and in-depth theoretical support for a wide range of empirically established network designs. Furthermore, it also offers a solid foundation to derive new deep learning approaches, with the additional help from the rich existing work on PGMs. It is thus a promising alternative neural network interpretation that deepens our theoretical understanding and unveils a new perspective for future deep learning research.

In the future, we plan to investigate a number of open questions that stem from this work, especially:

- Is the current chain graph interpretation sufficient to capture the full essence of neural networks? Based on the current results, we are reasonably optimistic that the proposed interpretation can cover an essential part of the neural network mechanism. However, compared to the function approximator view, it only covers a subset of existing techniques. Is this subset good enough?
- On a related note: can we find chain graph interpretations for other important network designs (or otherwise some chain graph interpretable alternatives with comparable or better performance)? The current work provides a good start, but it is by no means an exhaustive study.
- Finally, what other new deep learning models and procedures can we build up based on the chain graph framework? The partially collapsed feed-forward inference proposed in this work is just a simple illustrative example, and we believe that many other promising deep learning techniques can be derived from the proposed chain graph interpretation.

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

# A    GLOBALLY DIRECTED STRUCTURE OF LAYERED CHAIN GRAPH

With the absence of (undirected) intra-layer connection in the chain components $P(\mathbf{X}^l|Pa(\mathbf{X}^l))$, the layered chain graph defined in Definition 1 has a globally directed structure. This means that equivalently it can also be modeled by a directed graphical model (Bayesian network) which admits the same nodewise decomposition

$$P(\mathbf{X}^2, \ldots, \mathbf{X}^L|\mathbf{X}^1) = \prod_{l=2}^{L}\prod_{i=1}^{N^l} P(X_i^l|Pa(\mathbf{X}^l)) \tag{10}$$

and whose factorized nodewise conditional distributions $P(X_i^l|Pa(\mathbf{X}^l))$ are additionally modeled by pairwise CRFs:

$$P(X_i^l|Pa(\mathbf{X}^l)) = f^l\big(\mathbf{T}^l(X_i^l), \mathbf{e}_i^l\big(\mathbf{T}^{p_1}(\mathbf{X}^{p_1}), \ldots, \mathbf{T}^{p_n}(\mathbf{X}^{p_n})\big)\big). \tag{11}$$

The expressions Eq. (10), (11) are identical to Eq. (1), (2), which shows the equivalence. This is a rather straightforward result as directed graphical model is just a special case of chain graph. We employ the more general chain graph modeling in this paper out of two concerns:

1. Neural network designs rely heavily on the notion of layers. The layered chain graph formulation provides us with the notion of chain component that can correspond exactly to a neural network layer. This is missing in a directed graphical model formulation;
2. As we have discussed in Section 2.3, using the layered chain graph formulation allows for a straightforward extension from the classical neural network case (layered chain graph) to the more complex case with intra-connected layers that corresponds to general chain graphs with non-bipartite CRF chain components, which can be useful for, e.g., structured prediction tasks.

# B    PROOFS

## B.1    PROOF OF PROPOSITION 1

The main idea behind the proof is that for a linear function, its expectation can be moved inside and directly applied on its arguments. With this in mind let's start the actual deductions:

- To obtain Eq. (6), we start from Eqs. (5) and (2):

$$Q_i^l(x_i^l|\tilde{\mathbf{x}}^1) = \mathbb{E}_{Q^{p_1}, \ldots, Q^{p_n}}[P(x_i^l|Pa(\mathbf{X}^l))] \tag{12}$$

$$= \mathbb{E}_{Q^{p_1}, \ldots, Q^{p_n}}[f^l\big(\mathbf{T}^l(X_i^l), \mathbf{e}_i^l\big(\mathbf{T}^{p_1}(\mathbf{X}^{p_1}), \ldots, \mathbf{T}^{p_n}(\mathbf{X}^{p_n})\big)\big)]. \tag{13}$$

Now, we make the assumption that the following mapping is approximately linear:

$$(\mathbf{v}_1, \ldots, \mathbf{v}_n) \mapsto f^l\big(\mathbf{T}^l(X_i^l), \mathbf{e}_i^l(\mathbf{v}_1, \ldots, \mathbf{v}_n)\big). \tag{14}$$

This allows us to move the expectation inside, resulting in (racall the definition of $\mathbf{q}$ in Eq. (4))

$$Q_i^l(x_i^l|\tilde{\mathbf{x}}^1) \approx f^l\big(\mathbf{T}^l(X_i^l), \mathbf{e}_i^l\big(\mathbb{E}_{Q^{p_1}}[\mathbf{T}^{p_1}(\mathbf{X}^{p_1})], \ldots, \mathbb{E}_{Q^{p_n}}[\mathbf{T}^{p_n}(\mathbf{X}^{p_n})]\big)\big) \tag{15}$$

$$\approx f^l\big(\mathbf{T}^l(X_i^l), \mathbf{e}_i^l(\mathbf{q}^{p_1}, \ldots, \mathbf{q}^{p_n})\big). \tag{16}$$

- To obtain Eq. (7), we go through a similar procedure (for discrete RVs we replace integrations by summations):
  From Eqs. (4), (5) and (2) we have

$$\mathbf{q}_i^l = \mathbb{E}_{Q_i^l}[\mathbf{T}^l(X_i^l)] \tag{17}$$

$$= \int_{x_i^l} \mathbf{T}^l(x_i^l)Q_i^l(x_i^l|\tilde{\mathbf{x}}^1)dx_i^l \tag{18}$$

$$= \int_{x_i^l} \mathbf{T}^l(x_i^l)\mathbb{E}_{Q^{p_1}, \ldots, Q^{p_n}}[f^l\big(\mathbf{T}^l(x_i^l), \mathbf{e}_i^l\big(\mathbf{T}^{p_1}(\mathbf{X}^{p_1}), \ldots, \mathbf{T}^{p_n}(\mathbf{X}^{p_n})\big)\big)]dx_i^l \tag{19}$$

$$= \mathbb{E}_{Q^{p_1}, \ldots, Q^{p_n}}\Big[\int_{x_i^l} \mathbf{T}^l(x_i^l)f^l\big(\mathbf{T}^l(x_i^l), \mathbf{e}_i^l\big(\mathbf{T}^{p_1}(\mathbf{X}^{p_1}), \ldots, \mathbf{T}^{p_n}(\mathbf{X}^{p_n})\big)\big)dx_i^l\Big]. \tag{20}$$

Defining the activation function $\mathbf{g}^l$ as (thus $\mathbf{e}_i^l$ corresponds to the preactivation)

$$\mathbf{g}^l(\mathbf{v}) = \int_{x_i^l} \mathbf{T}^l(x_i^l) f^l\big(\mathbf{T}^l(x_i^l), \mathbf{v}\big) dx_i^l, \tag{21}$$

we then have

$$\mathbf{q}_i^l = \mathbb{E}_{Q^{p_1}, \ldots, Q^{p_n}} \big[\mathbf{g}^l\big(\mathbf{e}_i^l\big(\mathbf{T}^{p_1}(\mathbf{X}^{p_1}), \ldots, \mathbf{T}^{p_n}(\mathbf{X}^{p_n})\big)\big)\big]. \tag{22}$$

Again, we make another assumption that the following mapping is approximately linear:

$$(\mathbf{v}_1, \ldots, \mathbf{v}_n) \mapsto \mathbf{g}^l\big(\mathbf{e}_i^l(\mathbf{v}_1, \ldots, \mathbf{v}_n)\big). \tag{23}$$

This leads to the following approximation in a similar fashion:

$$\mathbf{q}_i^l \approx \mathbf{g}^l\big(\mathbf{e}_i^l\big(\mathbb{E}_{Q^{p_1}}[\mathbf{T}^{p_1}(\mathbf{X}^{p_1})], \ldots, \mathbb{E}_{Q^{p_n}}[\mathbf{T}^{p_n}(\mathbf{X}^{p_n})]\big)\big) \tag{24}$$

$$\approx \mathbf{g}^l\big(\mathbf{e}_i^l(\mathbf{q}^{p_1}, \ldots, \mathbf{q}^{p_n})\big). \tag{25}$$

$\square$

## B.2 PROOF (AND OTHER DETAILS) OF COROLLARY 2

Let's consider a node $X_i^l$ connected by parent layers $\mathbf{X}^{p_1}, \ldots, \mathbf{X}^{p_n}$. To lessen the notations we use the shorthands $\mathbf{e}_i^l$ for $\mathbf{e}_i^l\big(\mathbf{T}^{p_1}(\mathbf{X}^{p_1}), \ldots, \mathbf{T}^{p_n}(\mathbf{X}^{p_n})\big)$ and $\bar{\mathbf{e}}_i^l$ for $\mathbf{e}_i^l(\mathbf{q}^{p_1}, \ldots, \mathbf{q}^{p_n})$.

1. For the binary case we have

$$T^l(X_i^l) = X_i^l \in \{\alpha, \beta\}, \tag{26}$$

$$P(X_i^l|Pa(\mathbf{X}^l)) = f^l(X_i^l, e_i^l) = \frac{1}{Z(Pa(\mathbf{X}^l))} \exp(X_i^l \, e_i^l) \tag{27}$$

with the partition function

$$Z(Pa(\mathbf{X}^l)) = \exp(\alpha \, e_i^l) + \exp(\beta \, e_i^l) \tag{28}$$

that makes sure $P(X_i^l|Pa(\mathbf{X}^l))$ is normalized. This means that, since $X_i^l$ can either be $\alpha$ or $\beta$, we can equivalently write ($\sigma : x \mapsto 1/(1 + \exp(-x))$ denotes the sigmoid function)

$$f^l(X_i^l, e_i^l) = P(x_i^l|Pa(\mathbf{x}^l)) = \begin{cases} \sigma((\alpha - \beta) \, e_i^l) & \text{if } x_i^l = \alpha \\ \sigma((\beta - \alpha) \, e_i^l) & \text{if } x_i^l = \beta \end{cases} = \sigma((2x_i^l - \alpha - \beta) \, e_i^l). \tag{29}$$

Using the feed-forward expression (Eq. (7)), we have

$$q_i^l \approx \sum_{x_i^l \in \{\alpha, \beta\}} x_i^l \, f^l(x_i^l, \bar{e}_i^l) = \alpha \, \sigma((\alpha - \beta) \, \bar{e}_i^l) + \beta \, \sigma((\beta - \alpha) \, \bar{e}_i^l) \tag{30}$$

$$= \frac{\beta - \alpha}{2} \tanh\left(\frac{\beta - \alpha}{2} \cdot \bar{e}_i^l\right) + \frac{\alpha + \beta}{2}. \tag{31}$$

Especially,

$$\text{When } \alpha = 0, \beta = 1, \text{ we have } q_i^l \approx \sigma(\bar{e}_i^l); \tag{32}$$

$$\text{When } \alpha = -1, \beta = 1, \text{ we have } q_i^l \approx \tanh(\bar{e}_i^l). \tag{33}$$

Furthermore, the roles of $\alpha$ and $\beta$ are interchangeable.

2. For the multilabel case, let's assume that the node $X_i^l$ can take one of the $c$ labels $\{1, \ldots, c\}$. In this case, we have an indicator feature function which outputs a length-$c$ feature vector

$$\mathbf{T}^l(X_i^l) = (\mathbf{1}_{X_i^l=1}, \ldots, \mathbf{1}_{X_i^l=c})^\top. \tag{34}$$

This means that for any given label $j \in \{1, \ldots, c\}$, $\mathbf{T}^l(j)$ is a one-hot vector indicating the $j$-th position. Also, $\mathbf{e}_i^l$ and $\bar{\mathbf{e}}_i^l$ will both be vectors of length $c$, and we denote $e_{i,j}^l$ and $\bar{e}_{i,j}^l$ their $j$-th entries. We have then

$$P(X_i^l|Pa(\mathbf{X}^l)) = f^l(\mathbf{T}^l(X_i^l), \mathbf{e}_i^l) = \frac{1}{Z(Pa(\mathbf{X}^l))} \exp(\mathbf{T}^l(X_i^l) \cdot \mathbf{e}_i^l) \tag{35}$$

with the normalizer (i.e. partition function)

$$Z(Pa(\mathbf{X}^l)) = \sum_{j=1}^{c} \exp(e_{i,j}^l). \tag{36}$$

This means that

$$\forall j \in \{1, \ldots, c\}, f^l(\mathbf{T}^l(j), \mathbf{e}_i^l) = \big( \mathrm{softmax}(e_{i,1}^l, \ldots, e_{i,c}^l) \big)_j, \tag{37}$$

and, using the feed-forward expression (Eq. (7)), we have

$$\mathbf{q}_i^l \approx \sum_{x_i^l=1}^{c} f^l(\mathbf{T}^l(x_i^l), \bar{\mathbf{e}}_i^l)\mathbf{T}^l(x_i^l) = \sum_{j=1}^{c} \big( \mathrm{softmax}(\bar{e}_{i,1}^l, \ldots, \bar{e}_{i,c}^l) \big)_j \mathbf{T}^l(j) \tag{38}$$

$$= \mathrm{softmax}(\bar{e}_{i,1}^l, \ldots, \bar{e}_{i,c}^l), \tag{39}$$

i.e. the expected features $\mathbf{q}_i^l$ of the multi-labeled node $X_i^l$ is a length-$c$ vector that encodes the result of a $\mathrm{softmax}$ activation.

3. The analytical forms of the activation functions are quite complicated for rectified Gaussian nodes. Luckily, it is straight-forward to sample from rectified Gaussian distributions (get Gaussian samples, then rectify). meaning that we can easily evaluate them numerically with sample averages. A resulting visualization is displayed in Figure 1 Right. Specifically:

   - The ReLU nonlinearity can be approximated reasonably well by a rectified Gaussian node with no leak ($\epsilon = 0$) and $\tanh$-modulated standard deviation ($s_i^l = \tanh(e_i^l)$), as shown by the red plot in Figure 1 Right;
   - Similar to the ReLU case, the leaky ReLU nonlinearity can be approximated by a leaky ($\epsilon \neq 0$) rectified Gaussian node with $\tanh$-modulated standard deviation ($s_i^l = \tanh(e_i^l)$). See the green plot in Figure 1 Right which depict the case with leaky factor $\epsilon = 1/3$;
   - We discover that a rectified Gaussian node with no leak ($\epsilon = 0$) and an appropriately-chosen constant standard deviation $s_i^l$ can closely approximate the softplus nonlinearity (see the blue plot in Figure 1 Right). We numerically evaluate $s_i^l = 1.776091849725427$ to minimize the maximum pointwise approximation error.

   Averaging over more samples would of course lead to more accurate (visually thinner) plots, however in Figure 1 Right we deliberately only average over 200 samples, because we also want to visualize their stochastic behaviors: the perceived thickness of a plot can provide a hint to the output sample variance given the preactivation $e_i^l$.

$\square$

## B.3 PROOF OF PROPOSITION 3

Given a refinement module (c.f. Definition 2) that augments a base submodule $m$ from layer $\mathbf{X}^{l-1}$ to layer $\mathbf{X}^l$ using a refining submodule $r$ from $\tilde{\mathbf{X}}^l$ to layer $\mathbf{X}^l$, denote $\mathbf{g}^l$ the activation function corresponding to the distribution of nodes in layer $\mathbf{X}^l$, we assume that these two submodules alone would represent the following mappings during feed-forward

$$\begin{cases} \mathbf{q}^l = \mathbf{g}^l(\mathbf{e}^m(\mathbf{q}^{l-1})) & \text{(base submodule)} \\ \mathbf{q}^l = \mathbf{g}^l(\mathbf{e}^r(\tilde{\mathbf{q}}^l)) & \text{(refining submodule)} \end{cases} \tag{40}$$

where $\mathbf{e}^m$ and $\mathbf{e}^r$ represent the output preactivations of the base submodule and the refining submodule respectively. Then, given an input activation $\mathbf{q}^{l-1}$ from $\mathbf{X}^{l-1}$, the output preactivation of the overall refinement module should sum up contributions from both the main and the side branches (c.f. Eq. (3)), meaning that the refinement module computes the output as

$$\mathbf{q}^l = \mathbf{g}^l(\mathbf{e}^m(\mathbf{q}^{l-1}) + \mathbf{e}^r(\tilde{\mathbf{q}}^l)) \tag{41}$$

with $\tilde{\mathbf{q}}^l$ the output of the duplicated base submodule with shared weight, given by

$$\tilde{\mathbf{q}}^l = \mathbf{g}^l(\mathbf{e}^m(\mathbf{q}^{l-1})). \tag{42}$$

We have thus (Id denotes the identity function)

$$\mathbf{q}^l = \mathbf{g}^l \circ (\mathrm{Id} + \mathbf{e}^r \circ \mathbf{g}^l) \circ \mathbf{e}^m(\mathbf{q}^{l-1}) \tag{43}$$

where the function $\mathrm{Id} + \mathbf{e}^r \circ \mathbf{g}^l$ describes a preactivation residual block that arises naturally from the refinement module structure. $\square$

## B.4 Proof of Proposition 4

To match the typical deep learning formulations and ease the derivation, we assume that the base layered chain graph has a sequential structure, meaning that $Pa^t(\mathbf{X}^{l,t})$ contains only $\mathbf{X}^{l-1,t}$, and we have

$$P(X_i^{l,t}|Pa^t(\mathbf{X}^{l,t}), \mathbf{X}^{l,t-1}) = P(X_i^{l,t}|\mathbf{X}^{l-1,t}, \mathbf{X}^{l,t-1}) \tag{44}$$

$$= f^{l,t}\big(\mathbf{T}^l(X_i^{l,t}), \mathbf{e}_i^{l,t}\big(\mathbf{T}^{l-1}(\mathbf{X}^{l-1,t}), \mathbf{T}^l(\mathbf{X}^{l,t-1})\big)\big). \tag{45}$$

1. When the connection from $\mathbf{X}^{l,t-1}$ to $\mathbf{X}^{l,t}$ is dense, we have that for each $i \in \{1, \dots, N^l\}$,

$$\mathbf{e}_i^{l,t}\big(\mathbf{T}^{l-1}(\mathbf{X}^{l-1,t}), \mathbf{T}^l(\mathbf{X}^{l,t-1})\big) = \sum_{j=1}^{N^{l-1}} \mathbf{W}_{j,i}^l \mathbf{T}^{l-1}(X_j^{l-1,t}) + \sum_{k=1}^{N^l} \mathbf{U}_{k,i}^l \mathbf{T}^l(X_k^{l,t-1}) + \mathbf{b}_i^l. \tag{46}$$

   Thus the feed-forward update for layer $l$ at time $t$ is

$$\forall i \in \{1, \dots, N^l\}, \ \mathbf{q}_i^{l,t} \approx \mathbf{g}^l\Big( \sum_{j=1}^{N^{l-1}} \mathbf{W}_{j,i}^l \mathbf{q}_j^{l-1,t} + \sum_{k=1}^{N^l} \mathbf{U}_{k,i}^l \mathbf{q}_k^{l,t-1} + \mathbf{b}_i^l \Big) \tag{47}$$

   which corresponds to the update of a simple recurrent layer.
2. The assumption of intra-layer conditional independence through time means that we have

$$P(X_i^{l,t}|Pa^t(\mathbf{X}^{l,t}), \mathbf{X}^{l,t-1}) = P(X_i^{l,t}|\mathbf{X}^{l-1,t}, X_i^{l,t-1}), \tag{48}$$

   which in terms of preactivation function means that for each $i \in \{1, \dots, N^l\}$,

$$\mathbf{e}_i^{l,t}\big(\mathbf{T}^{l-1}(\mathbf{X}^{l-1,t}), \mathbf{T}^l(\mathbf{X}^{l,t-1})\big) = \mathbf{e}_i^{l,t}\big(\mathbf{T}^{l-1}(\mathbf{X}^{l-1,t}), \mathbf{T}^l(X_i^{l,t-1})\big) \tag{49}$$

$$= \sum_{j=1}^{N^{l-1}} \mathbf{W}_{j,i}^l \mathbf{T}^{l-1}(X_j^{l-1,t}) + \mathbf{U}_i^l \mathbf{T}^l(X_i^{l,t-1}) + \mathbf{b}_i^l. \tag{50}$$

   In this case the feed-forward update for layer $l$ at time $t$ is

$$\forall i \in \{1, \dots, N^l\}, \ \mathbf{q}_i^{l,t} \approx \mathbf{g}^l\Big( \sum_{j=1}^{N^{l-1}} \mathbf{W}_{j,i}^l \mathbf{q}_j^{l-1,t} + \mathbf{U}_i^l \mathbf{q}_i^{l,t-1} + \mathbf{b}_i^l \Big) \tag{51}$$

   which corresponds to the update of an independently RNN layer (c.f. Eq. (2) of Li et al. (2018)).

   $\square$

## B.5 Proof of Proposition 5

Again, to match the typical deep learning formulations and ease the derivation, we assume that the layered chain graph has a sequential structure, meaning that $\mathbf{X}^l$ is only the parent layer of $\mathbf{X}^{l+1}$. With the introduction of the auxiliary Bernoulli RVs $\mathbf{D}^l$, the $l + 1$-th chain component represents

$$P(\mathbf{X}^{l+1}|\mathbf{X}^l, \mathbf{D}^l) = \prod_{j=1}^{N^{l+1}} f_j^{l+1}\big(\mathbf{T}^{l+1}(X_j^{l+1}), \mathbf{e}_j^{l+1}\big(\mathbf{T}^l(\mathbf{X}^l), \mathbf{D}^l\big)\big) \tag{52}$$

with

$$\mathbf{e}_j^{l+1}\big(\mathbf{T}^l(\mathbf{X}^l), \mathbf{D}^l\big) = \sum_{i=1}^{N^{l-1}} D_i^l \mathbf{W}_{i,j}^{l+1} \mathbf{T}^l(X_i^l) + \mathbf{b}_j^l. \tag{53}$$

- For a feed-forward pass during training, we draw a sample $d_i^l$ for each Bernoulli RV $D_i^l$, and the feed-forward update for layer $l + 1$ becomes

$$\forall j \in \{1, \dots, N^{l+1}\}, \ \mathbf{q}_j^{l+1} \approx \mathbf{g}^{l+1}\Big( \sum_{i=1}^{N^l} d_i^l \mathbf{W}_{i,j}^{l+1} \mathbf{q}_i^l + \mathbf{b}_j^l \Big). \tag{54}$$

  Since $d_i^l = 1$ with probability $p^l$ and $d_i^l = 0$ with probability $1 - p^l$, each activation $\mathbf{q}_i^l$ will be "dropped out" (i.e. $d_i^l \mathbf{q}_i^l = 0$) with probability $1 - p^l$ (this affects all $\mathbf{q}_j^{l+1}$ simultaneously).

- For a feed-forward pass at test time, we marginalize each Bernoulli RV $D_i^l$. Since we have

$$\forall i \in \{1, \ldots, N^l\}, \ \mathbb{E}[D_i^l] = p^l, \tag{55}$$

the feed-forward update for layer $l + 1$ in this case becomes

$$\forall j \in \{1, \ldots, N^{l+1}\}, \ \mathbf{q}_j^{l+1} \approx \mathbf{g}^{l+1}\big(\sum_{i=1}^{N^l} p^l \mathbf{W}_{i,j}^{l+1} \mathbf{q}_i^l + \mathbf{b}_j^l\big) \tag{56}$$

where we see the appearance of the constant scale $p^l$.

$\square$

## C  REMARK ON INPUT MODELING

A technical detail that was not elaborated in the discussion of chain graph interpretation (Section 2) is the input modeling: How to encode an input data sample? Ordinarily, an input data sample is treated as a sample drawn from some distribution that represents the input. In our case however, since the feed-forward process only pass through feature expectations, we can also directly interpret an input data sample as a feature expectation, meaning as a resulting average rather than a single sample. Using this fact, Shen et al. (2019) propose the "soft clamping" approach to encode real valued input taken from an interval, such as pixel intensity, simply as an expected value of a binary node which chooses between the interval boundary values.

This said, since only the conditional distribution $P(\mathbf{X}^2, \ldots, \mathbf{X}^L | \mathbf{X}^1)$ is modeled, our discriminative setting actually do not require specifying an input distribution $P(\mathbf{X}^1)$.

