# OpenReview forum: "A Chain Graph Interpretation of Real-World Neural Networks"
_ICLR.cc/2021/Conference — Reject_

### Official Review · AnonReviewer2 · 2020-10-26
**The paper fails to deliver on its promises**

**Rating:** 3
**Confidence:** 4

**Review:**

###############################################################################

Summary

Summarize what the paper claims to contribute. Be positive and generous.

The authors propose an interpretation of feed-forward neural networks and recursive neural networks as chain graphs (CGs). They claim this new interpretation can provide novel theoretical support an insights to existing techniques, as well allow for the derivation of new approaches.

###############################################################################

Pros and cons

Pros:
 - the text is well-written
 - the attempt at studying neural networks under a graphical probabilistic model perspective is praisable

Cons:
 - the chain graph interpretation put forward by the authors is superfluous, as in reality the definition found in the apper is that of DAGs with particular parametric constraints
 - most of the results presented by the authors do not rely on the CG interpretation, and can already be found in the litterature
 - the paper is missing a connexion to stochastic feedforward neural networks (SFNNs), to which the proposed interpretation is extremely similar

###############################################################################

Recommendation

I recommend rejection of the paper, for two reasons. First, I believe most of the contributions proposed in the paper are not novel, as they can already be found in published form. See my detailed comments below. Second, the CG interpretation put forward by the authors is trivial and superfluous, since the authors in reality only consider CGs restricted to DAGs. As such, the whole claim of the paper that CGs can give a new, relevant perspective to NNs, is not credible.

###############################################################################

Questions to authors

I would like the authors to comment on the fact that their CG interpretation is indeed the LWF-CG interpretation, restricted to DAGs (singleton chain components). Furthermore, I would appreciate if the authors could relate their approach to the SFNN model, and what differs from that interpretation.

###############################################################################

Detailed comments:

p.1 §3: an efficient approximate probabilistic inference -> What is meant here by efficient ? Is it an unbiased estimator ? What is meant by inference ? Computing $p(y|x)$ ? $\arg \max_y p(y|x)$ ?

p.2 §2: The chain graph model -> There exists at least 4 chain graph interpretations. See
Drton, Mathias. „Discrete Chain Graph Models“. In: Bernoulli 15 (Sept. 2009),
pp. 736–753.
Among those, there are two which subsume UGs and DAGs, while the two others subsume BGs and DAGs. If you follow the CG interpretation from Koller and Friedman then you assume NNs are LWF-CGs.
I strongly suggest that you make it explicit which chain graph interpretation you follow, for the sake of clarity. These interpretations are not equivalent.

p.2 §2: there exists a series of works [...] -> I believe the list is much bigger than that. You forget very popular models which are direct instanciation of PGMs, such as VAEs, HMMs, LDAs, GMMs, CRFs, and all of their variants.

p.2 §4: an approximate probabilistic inference procedure -> What is meant by that ?

p.2 §4: provides additional insights -> such as ?

p.3 §2: layered chain graph -> A chain graph is always layered into chain components. The name "layered chain graph" is confusing, as it seems to imply that some chain graphs may not be layered. Since you are using the LWF-CG interpretation, you might point to the relevant papers where the graphical model was introduced, and simply re-use the establihsed LWF-CG name from the litterature, instead of inventing a new one.
Lauritzen, Steffen L. and Wermuth, N. „Graphical Models for Associations
between Variables, some of which are Qualitative and some Quantitative“. In:
The Annals of Statistics 17.1 (Mar. 1989), pp. 31–57
Frydenberg, M. (1990). The chain graph Markov property. Scand. J. Statist. 17 333–353. MR1096723

p.3 eq.1: This factorization is not that of a CG, but that of a DAG. Since you assume no undirected connection between variables in the same layer, then all chain component in your chain graph actually contain a single variable. What is the point of introducing the conecpt of CGs then, if you only consider CGs which are restricted to DAGs ?

p.3 eq.2: The LWF-CG interpretation, which you seem to follow since you refer to Koller and Friedman, does not imply this factorization for the graphs you consider. Since each chain component $K$ in your graph is a singleton, each $X_i$ and its parents form a clique in the closure graph $K$. As such, the CG structure does not imply any further factorization for p(x|pa(x))... It does seem your interpretation of NNs is not as general CGs, but as very specific DAGs with parametric constraints. Furthermore, you do not define what are $T$ and $f$ here.

p.3 Section 2.2: I believe your CG interpretation is simply a reformulation of stochastic neural networks, for which there already exists a great body of work. See, eg:

Eric Jang Shixiang Gu Ben Poole. Categorical Reparameterization with Gumbel-Softmax. ICLR (2017)

Yoshua Bengio, Nicholas Leonard, and Aaron Courville. Estimating or propagating gradients through stochastic neurons for conditional computation. arXiv preprint arXiv:1308.3432, 2013.

Yichuan Tang, Ruslan Salakhutdinov. Learning Stochastic Feedforward Neural Networks. NIPS 2013

p.4 Proposition 2.1: This result seems very limited, since it assumes linear layers, and in the end a linear NN.

p.4 Corollary 2: What are alpha and beta here ? How do they relate to $f$, $e$ or $T$ from Definition 1 ? What are $e_i^l$ and $s_i^l$ here ?

p.4 §6: the modularity of chain components justifies transfer learning via partial reuse of pre-trained networks -> I do not understand this argument. What is meant here by "modularity" ?

p.4 §7: However, feed-forward is no longer applicable through these intra-connected layers. -> You finally give an example here that corresponds to non-trivial chain graphs (with chain components greater than 1), and recognize that your theoretical results do not apply any more. This contradicts your claim that CGs offer a general interpretation of NNs with theoretical support. I believe the CG interpretation is not justified, nor required to derive the results you present in this paper.

p.5 Definition 2: I fail to see the point of introducing this new concept, which is a re-definition of residual blocks.

p.5 §4: While a vanilla layered [...] -> I fail to understand where the CG interpretation fits in this argument.

p.5 Proposition 4: This result seems trivial to me, and does not require the concept of a CG. Once you show that a linear layer followed by a specific activation can be interpreted as a parametric model for $p(\textit{out}|\textit{in})$, then any NN that unrolls as an acyclic directed graph can be interpreted as a probabilistic model.

p.6 §1: The simple recurrent layer, [...] -> Again, I do not see where the CG interpretation gives any insight here.

p.6 Section 3.3: This result, again, does not require the CG interpretation. It is also already know. See, e.g., Pierre Baldi and Peter J. Sadowski. Understanding Dropout. NIPS 2013

p.6 Section 4: It seems to me you are reinventing SFFNs. See, Yichuan Tang and Ruslan Salakhutdinov. Stochastic Feedforward Neural Network. NIPS 2013

p.7 §2: the sampling operation is not differentiable -> This is not true. See, e.g.,  the reparameterization trick for VAEs. You mention this in the next sentence...

---

> ### Author Response · Authors · 2020-11-16
> **Response Part 3 of 3**
>
> p.3 eq.2: We have already clarified the layered chain graph representation in the main response part. To answer your questions on what T and f are: we have already stated (p.3 §2) that T is a feature function (i.e., a statistic). As defined in Eq.2, f can be an arbitrary function that corresponds to a probability distribution. We have also added this comment on f in the revised version. Thank you for pointing this out.
>
> p.3 Section 2.2: Our CG interpretation is clearly not a reformulation of stochastic neural networks. After all, the goal of Section 2.2 is to interpret feed-forward, which is a deterministic process in its basic form. This said, we appreciate your curated list on the stochastic NNs that unfortunately accompanies your inaccurate claim.
>
> p.4 Proposition 2.1: While we request linear approximation, the end result after this approximation is non-linear, as stated in the proposition. It should also be clear from the proof of this proposition and the discussion in Corollary 2 that we are interpreting practical NNs with non-linear activations. You are welcome to look at the proof which rigorously derived how this is achieved, and we are at your disposal if you have more questions.
>
> p.4 Corollary 2: As we have stated in the paper, alpha and beta are the two possible values a binary node can take. Here we generalize over the Bernoulli nodes where typically the two labels are 0 and 1; e_i^l is the preactivation that is already defined in Eq.3, and s_i^l is the standard deviation of the Gaussian distribution that Y_i^l follows and can depend on e_i^l. Due to space constraint, we have put the detailed derivations in the appendix. We welcome you to have a closer look at it, since there we clearly specified what "f" and "T" are in each case.
>
> p.4 §6: "Modularity" here means that a layered chain graph, along with its trainable parameters, can be factorized into chain components that are CRFs. Especially, unlike undirected graphical models such as Boltzmann machines, we do not have a global normalizing term (i.e., partition function) that entangles all the parameters. Instead, all parameters are confined within their respective chain components and can thus be transferred in parts.
>
> p.4 §7: There is no contradiction here. We state clearly in our paper that feed-forward applies to layered chain graph, and here we state the fact that we can extend layered chain graph to general chain graph with non-bipartite CRF chain components, but feed-forward would no longer apply for these intra-connected layers (i.e., non-bipartite CRF chain components).
>
> p.5 Definition 2: This is not a re-definition of residual blocks. With this definition, we clarify the terms we are using to derive the chain graph interpretation of preactivation residual blocks.
>
> p.7 §2: The sampling operation by itself is not directly differentiable. And this is exactly why methods like reparametrization trick or Gumbel softmax trick are proposed. We have clearly discussed it in the paper and do not see any issue here.
>
>
> Hopefully our response has addressed all your concerns and you would agree that contrary to your initial claim, this paper does deliver on its promises. We would like to urge you to reconsider the merit of this paper. Please feel free to reach out to us for more questions and comments. We are at your disposal for further discussions.
>
>
> [1] Zheng, Shuai, et al. "Conditional random fields as recurrent neural networks." ICCV 2015
> [2] Baldi, Pierre, and Peter J. Sadowski. "Understanding dropout." NeurIPS 2013
> [3] Tang, Charlie, and Russ R. Salakhutdinov. "Learning stochastic feedforward neural networks." NeurIPS 2013
> [4] Nowozin, Sebastian, and Christoph H. Lampert. Structured learning and prediction in computer vision. Now publishers Inc, 2011.
> [5] Koller, Daphne, and Nir Friedman. Probabilistic graphical models: principles and techniques. MIT press, 2009

---

> ### Author Response · Authors · 2020-11-16
> **Response Part 2 of 3**
>
> To address the third point in "cons": Our work clearly differs from that of SFNN [3]. Apart from the straightforward fact that PCFF uses forward sampling (ancestral sampling) and SFNN uses importance sampling, SFNN does not provide a PGM interpretation to NNs, and indeed their approach clearly distinguishes between the stochastic units (random variables, PGM part) and the deterministic units (neurons, NN part). Our work provides a PGM interpretation to NNs so that each neuron IS a random variable, and it can switch between stochastic (ancestral sampling) and deterministic (feed-forward) behaviors depending on the inference method. This is also the main observation that leads to the PCFF method proposed in our work, where at each forward pass a different set of nodes have stochastic behavior, contrary to the SFNN case. Following your request, we have added additional discussions in related work (Section 1.1) and also in Section 4 to make sure the relations and differences w.r.t. SFNN are clarified. To summarize, it is clear that the statement of the proposed interpretation being "extremely similar" to SFNN is inaccurate. This should also address your detailed comments p.6 Section 4 and partially p.3 section 2.2.
>
>
> To address the rest of your detailed comments:
>
> p.1 §3 (+ p.2 §4 (1)): Efficient means that feed-forward is fast, especially faster than other traditional approximate probabilistic inference methods, including variational inference methods like mean field, LBP, etc and MCMC sampling methods, to name a few (of course, the exact probabilistic inference in this case is intractable); feed-forward outputs deterministic and biased estimations; "probabilistic inference" is a standard term [4] that signifies we are trying to compute the marginal distributions (p(y|x)) and not doing the MAP inference (which computes argmax_y p(y|x)) or MPE inference. All this should be clear from the later discussions in this paper.
>
> p.2 §2 (1): We agree with you that there are different chain graph variants. That is why we clearly cite the Koller and Friedman book [5], a standard reference for PGM, to specify that we are following the definition in this book (which is, as you have pointed out, the LWF-CG. [5] has also referenced the Frydenberg paper.). To make sure that there is no room for confusion, we have added the citations that you have provided in (p.3 §2) in the revised version of the paper. We thank you for providing the references.
>
> p.2 §2 (2): In the list that you have provided, VAE is not a considered a graphical model in the common sense (unless you count the interpretation of this work), HMMs, LDAs, GMMs, CRFs in their basic forms are all rather simple and shallow graphical models. Especially, basic HMMs and CRFs do not contain hidden variables. Of course you can build up hierarchical variants, and in fact, the layered chain graph in our paper is a hierarchical GM that is composed of CRFs.
>
> p.2 §4 (2): Please refer to Section 3 of our paper.

---

> ### Author Response · Authors · 2020-11-16
> **Response Part 1 of 3**
>
> Thank you for your time and detailed comments. It is nice to meet a reviewer who knows there are different variants of chain graphs.
>
>
> Before other things, since you are doubting our credibility and claiming that we "superfluously" employed the notion of chain graph (or LWF-CG to be more precise), we would like to first assure you of our good faith with the fact that the decision of using chain graph modeling is not made superfluously to impress readers with some fancy notions, but rather that we find it to be the appropriate choice. First of all, we do agree with you that the special PGM construction that corresponds to classical NNs, the one we refer to as "layered chain graph" to indicate that it is a specific chain graph construct, has indeed a globally directed structure (DAG) and is in fact also a directed graphical model (Bayesian network) where all its nodewise conditional distributions from its nodewise factorization are additionally constrained to be pairwise CRFs. This does not contradict with the fact that it can be viewed as a chain graph, which is a just strict generalization. We decided after careful consideration to use the more general chain graph modeling for the following two reasons:
>
> 1. NN designs rely heavily on the notion of layers. The chain graph formulation provides us with the notion of chain component that can correspond exactly to a layer in a NN (hence the name "layered chain graph"). This is missing in a directed graphical model (Bayesian network) formulation.
>
> 2. Using the more general chain graph formulation allows for a straightforward extension from the classical NN case (layered chain graph, with bipartite CRF chain components) to the more complex case with intra-connected layers that corresponds to general (LWF-)CGs with non-bipartite CRF chain components, e.g., the output layer of CRFasRNN [1], as we have already explained in the paper.
>
> To be a hundred percent sure this issue is addressed properly, we have added a section in the appendix in the revised version to comment on this point in detail. Hopefully this would help you realize that your criticism (first point in "cons" and second argument for rejection recommendation) is ill-founded. This should also address your detailed comments p.3 §2, p.3 eq.1 and partially p.3 eq.2.
>
>
> To address the first argument for rejection recommendation and the second point in "cons": The main focus of this paper is not on inventing a new variant of deep learning, but to provide a PGM-based theoretical interpretation to real-world NNs that people uses in practice. Thus it is not surprising that we need to discuss many existing works that are found in published form in the literature. We state clearly in the paper that the main contribution is to point out the exact correspondence between layered chain graphs and NNs, and we never claimed that the existing works themselves are our contributions. To show that this interpretation is also helpful for finding new approaches, we also proposed in Section 4 a novel method (PCFF) based on the layered CG interpretation. To answer your claim that "most of the results presented by the authors do not rely on the CG interpretation": If not mistaken, you are referring to the discussions in Section 3, because Sections 2 and 4 clearly rely on the CG interpretation. The goal of Section 3 is to show that the chain graph interpretation enables a holistic view that provides consistent theoretical support to various NN designs, and not that it is particularly indispensable for the theoretical analysis of any specific design (and we never claim this), nor that other forms of theoretical analyses are invalid. Of course, since all probability distributions are functions, one can always argue that all results can be formed from the function approximator view, however this is not as informative from the theoretical standpoint. To conclude, the response in this paragraph should clearly show that this criticism on the novelty of this paper is inappropriate. This should also address your detailed comments p.5 §4, p.5 proposition 4, p.6 §1 and p.6 Section 3.3 (The work you referred to [2] only mentions the "underlying acyclic graph" once in the linear case and still follows the function approximator view when analyzing the non-linear case that correspond to the real world NNs).

---

### Official Review · AnonReviewer3 · 2020-10-28
**alternative interpretation based on chain graph**

**Rating:** 4
**Confidence:** 3

**Review:**

This paper tries to interpret neural networks with chain graphs that provides theoretical analysis on various neural network components. Furthermore, this chain graph interpretation has been used to propose a new approach (architecture), which is a partially collapsed feed-forward. A layered chain graph representation is adopted to formulate the neural networks with layered chain graphs. This further establishes to interpret feed-forward as an approximate probabilistic inference with using linear approximations. Some concrete examples are shown to be analyzed  based on the chain graph formulation.

The overall context (analysis) seems straightforward to interpret the neural networks with chain graphs, but it is hard to achieve some meaningful information from this new interpretation to improve the current neural network models in terms of learning procedure or optimization.  The proposed partially collapsed feed-forward is a good example to come up with a new approach based on the chain graph interpretation. However, in terms of performance and complexity, it is practically not showing impressive improvements compared to the baseline methods. Moreover, it seems quite similar to previous works as far as I remember and one similar work is 'stochastic feedforward neural networks'. I fully agree the future works (open questions) in the conclusion and discussion section that this work still needs more investigations although this paper is a good initiative work.

---

> ### Author Response · Authors · 2020-11-16
> **Response Part 2 of 2**
>
> Regarding the third point, the proposed PCFF method is clearly different from SFNN. To start with, it is quite apparent that they are based on different sampling techniques (PCFF uses ancestral sampling whereas SFNN uses importance sampling). Moreover, in SFNN the authors clearly distinguishes the roles of stochastic units (random variables, PGM part) and the deterministic units (neurons, NN part), whereas PCFF is based on the CG interpretation of NNs which identifies neurons with random variables. Thus the nodes can freely switch between stochastic (ancestral sampling) and deterministic (feed-forward) behaviors depending on how inference is performed, and no distinction is needed. The PCFF method randomly chooses a different set of nodes to sample during each forward pass, contrary to the SFNN case where the stochastic part is always fixed. To make sure that all these are clarified, we have added additional discussions in related work (Section 1.1) and also in Section 4 in the revised version of the paper.
>
>
> We are glad that you find this paper to be "a good initiative work", however we beg to differ that "this work still needs more investigations", since this would mean that the main results of this paper would only be preliminary, which is not the case. As a theory-focused paper, it has successfully fulfilled its goal of establishing the exact correspondence between NNs and layered CGs. Moreover, we have provided sufficient concrete examples to demonstrate the usefulness of this CG interpretation on providing coherent theoretical support for existing methods and on discovering new approaches (PCFF). Investigating in depth what possible algorithms/applications this interpretation can bring would deviate from the focus of the current theoretical analysis and would also be unrealistic to tackle additionally in this rather short conference paper. This is why it is more appropriate to leave these goals for future work. (We do believe these goals are important and we are already working on them in our subsequent projects.)
>
>
> Hopefully our response has cleared up your concerns and you would agree that this paper deserves better merit. Please feel free to reach out to us for more questions and comments. We are at your disposal for further discussions.
>
>
> [1] Tang, Charlie, and Russ R. Salakhutdinov. "Learning stochastic feedforward neural networks." NeurIPS 2013

---

> ### Author Response · Authors · 2020-11-16
> **Response Part 1 of 2**
>
> Thank you for your time and constructive feedback!
>
>
> If we understand you correctly, the negative rating you provided is mainly based on the following three concerns:
>
> 1) You have the impression that the analysis is straightforward, but hard to achieve meaningful information and improvement on NNs;
>
> 2) You find that PCFF is not showing impressive improvements;
>
> 3) It seems to you that PCFF is quite similar to stochastic feedforward neural networks (SFNNs) [1].
>
> We would like to address these concerns in the following paragraphs and help you clarify the merit of this paper.
>
>
> Regarding the first point, while we understand why you might have this impression, it is not really the case. As this is a theory-focused paper where the main goal is to establish the exact correspondence between NNs and layered CGs, you might be unsatisfied if you are seeking concrete new model designs or recipes to improve the empirical performance of NNs, however this is not the focus of the current work. Despite this, the proposed chain graph interpretation is certainly useful for improving the current neural network models and achieve meaningful information. We would like to refer you to the 4th paragraph in our response to reviewer 1, who also raises a similar issue on the usefulness of our work. We believe that our answer there can also respond to your concern. Also, we are happy that you find the analysis to be straightforward, since this would hopefully mean that we have clearly explained the main ideas. This said, the main contribution of this paper, i.e., establishing the exact correspondence between NNs and layered CGs, is by no means a trivial or insignificant matter. We have been working on the combination of PGMs and NNs for quite some time, and figuring out this exact correspondence has been an exciting revelation, since this would really allow us to combine the best of both DL and PGM worlds. Reviewer 4 also remarks that "This paper provides an interesting result to bridge the two previously unrelated fields, which equips the community with useful insights to spark new research directions".
>
>
> Regarding the second point, PCFF is only proposed as a simple proof-of-concept example to demonstrate that the proposed chain graph interpretation (which is the main contribution of this paper) can be helpful for discovering interesting new approaches. Fixating on its performance when judging the merit of this theoretical paper would be off-target and missing the true value of this paper. This said, while PCFF does not impressively improve on the performance compared to dropout, it is significantly better than dropout in terms of consistency on improving the results. As made especially clear in the experiment on CIFAR10 image classification task with resnet (See Figure 4 Right), an inappropriate drop rate for dropout can significantly deteriorate the performance below the unregularized baseline, whereas PCFF is less sensitive to the hyperparameter choice (sample rate) and delivers significantly more consistent improvement (and degrades more gracefully when the variance is too high).

---

### Official Review · AnonReviewer1 · 2020-10-29
**Promising idea but results are too simple and not adequately presented.**

**Rating:** 4
**Confidence:** 4

**Review:**

The authors look for ways to frame neural networks in terms of probabilistic models. This is a worthy goal and I believe the authors should pursue this path with enthusiasm. The theme is relevant, however the novel ideas that are in the paper seem to have relatively small significance. The translation from neural networks to chain graphs is not surprising; actually if I understand the translation we basically get a Bayesian network out of the directed acyclic graph encoding the connections in a neural network (I must say that the translation could be presented in a more didactic fashion. In any case, given this translation, the authors are able to frame some techniques from neural networks as techniques from chain graphs. But this feels too forced a translation, because neural networks do have a natural interpretation and usage as function approximators; why should one go all this ways to find a probabilistic translation that explains a few well known things, and opens the possibility of some probabilistic algorithms that could anyway be derived in the context of neural networks? One example: given the proposed translation, it is obvious that sequential models will appear as dynamic chain graphs, why is this useful in any way?

On top of this, I find it troublesome that the authors "prove" results by saying things like "approximately linear" and "approximately" this and that. How can these results be proven given this level of informal justification. I really think this should be fixed before publication.

Some sentences are a bit confusing. In Page 3, for instance, the authors say that a model with P(X|Pa(X)) is modeled by a CRF; usually a CRF models a discriminative model, is this the case here? What exactly is going on? Also the last paragraph of Section 3.2 is very hard to parse (the main point there is not clear at all).

---

> ### Author Response · Authors · 2020-11-16
> **Response Part 2 of 2**
>
> To answer your specific questions:
>
> - Why "proving approximations"?
>
> If not mistaken, you are specifically referring to Proposition 1 in the paper. The fact that we employ linear approximations to obtain the results doesn't mean that the reasoning is "informal" or "troublesome" in any way. While we try to state the main results in the paper in a concise and human-readable manner, we always accompany each of them with a formal proof to rigorously formulate the the problem and derive the results. This is also the case for Proposition 1 where in the proof we explained rigorously how the linear approximation is carried out and how the results are derived. Thus we do think we have adequately presented the results in this paper. Due to space constraints we have put all the proofs in the appendix. However we welcome you to have a closer look at them, and we are happy to answer all questions you might have.
>
> - What is going on with "CRF"?
>
> By definition, a conditional random field (CRF) is an undirected graphical model (i.e. Markov random field) that models a conditional distribution. This is exactly what we are referring to. Since CRF models conditional distributions instead of joint distributions, it is indeed a discriminative model. There is no contradiction here.
>
> - The main point of 3.2?
>
> We are not sure why you find the last paragraph of Section 3.2 very hard to parse. To summarize the main point of this paragraph: here we state that in practice, IndRNN has been shown to work better than other variants including vanilla RNN and LSTM, and we hypothesize that the intra-layer conditional independence through time assumption, which is made apparent from the dynamic chain graph interpretation of IndRNN, might be the differentiating factor that leads to the empirical success of IndRNN, and that this assumption could be particularly suited for sequential modeling.
>
>
> Hopefully our response has addressed your concerns and you would reconsider the merit of this paper. We hope that you would agree the statement "results are too simple and not adequately presented" is not well justified. Please feel free to reach out to us for more questions and comments. We are at your disposal for further discussions.
>
>
> [1] Lee, Jaehoon, et al. "Deep neural networks as gaussian processes." ICLR 2018

---

> ### Author Response · Authors · 2020-11-16
> **Response Part 1 of 2**
>
> Thank you for your time and constructive feedback!
>
>
> We are glad that you also agree we are working on a worthy goal, however we respectfully disagree that the work in this paper has "relatively small significance". The end result might look simple at the first glance (hopefully this also means we have clearly explained the main ideas), but the main result of this paper, the exact correspondence between NNs and layered chain graphs, is by no means a trivial result or insignificant. It provides significantly stronger theoretical support than the generic "function approximator" view, and equips the NN components with concrete probabilistic semantics, clarifying the modeling assumptions and approximations. As we have demonstrated in the paper, this chain graph interpretation can provide coherent and in-depth theoretical support for various practical NN designs, and is also helpful for discovering new approaches. We have been wondering about the relation between PGMs and NNs for quite some time, and figuring out this exact correspondence has been a surprising and exciting revelation for us. We believe that this work presents an important theoretical breakthrough, and, as reviewer 4 says, "provides an interesting result to bridge the two previously unrelated fields, which equips the community with useful insights to spark new research directions".
>
>
> Also, we beg to differ that the chain graph interpretation is "too forced a translation". On the contrary, the proposed chain graph interpretation can model real world NNs in a clear and natural way, without any "forced" conditions or nonsensical conclusions that contradict the empirical behaviors of NNs used in practice, especially compare to prior work such as [1] which rely on unrealistic assumptions like infinite width layers.
>
>
> In terms of usefulness, it should be clear that this paper focuses on theoretical analysis of NNs and not on designing concrete networks to achieve best results on specific tasks. Hopefully you would agree with us that theoretical advances, meaning better understanding of the underlying mechanism of NNs, is as important as improving the empirical performance, if not even more. Moreover, they should be judged with different metrics, and theoretical results from papers like this one shouldn't be criticized as "too simple" or useless because they do not provide a concrete recipe to beat some benchmarks for specific tasks. While we agree that with endless trial-and-error, one would eventually find any algorithm that works on NNs, the whole point of developing better theory is to benefit from a more rigorous theoretical understanding and conduct more principled research, so as to make faster progress with less "random walks" or "alchemy". Taking the sequential modeling example you mentioned (Section 3.2), the clear modeling of dynamic chain graph gives us a rigorous theoretical basis to raise and tackle questions such as: Would relaxing the Markov assumption made by RNNs lead to stronger models? How can one perform uncertainty estimation with RNNs? Can the assumption of intra-layer conditional independence through time be applied and be beneficial to other types of existing RNNs? Pursuing any of these directions might lead to a series of interesting results. But of course they are out of the scope of the current work, which focuses on showing the correspondence between NNs and layered chain graphs, and it is also unrealistic to include them all within the 8 page limit. These would therefore be more suited for future work.

---

### Official Review · AnonReviewer4 · 2020-10-29
**An interesting paper**

**Rating:** 6
**Confidence:** 2

**Review:**

Update: after reading the feedback and discussing with the other reviewers, I decide to keep my score unchanged.

Original comments:
In this paper, the authors provide new interpretation of neural networks via chain graphs, which can be used as a new theoretical framework to understand the behavior of neural networks.

Pros.

1. Although both PMG and Neural networks are based on graphs, very little research has been proposed to bring the two fields together. This paper provides an interesting result to bridge the two previously unrelated fields, which equips the community with useful insights to spark new research directions.

2. This paper helps the community to fully utilize the existing works from PGM to solve current obstacles in deep learning.

3. The open questions are also interesting. I would be particular interested in the third question.

Overall, I think this is an interesting paper.

---

> ### Author Response · Authors · 2020-11-16
> **Response**
>
> Thank you for your time and your positive feedback!
>
> We are really glad that you find our paper interesting and are grateful for your support. We are at your disposal if you wish for further discussions.

---

### Author Response · Authors · 2020-11-16
**Summary of changes in the revised version of the paper**

- Clarify that the chain graph model is also referred to as LWF chain graph model in some litteratures.
- Explain the relation and differencies of this work compared to stochastic feedforward neural network.
- Explain that $f^l$ in Eq.2 is a generic function that represents a probability distribution.
- Detail the equivalence between layered chain graph and its corresponding Bayesian network representation, and why it makes sense to use the chain graph modeling.

---

### Decision · Program_Chairs · 2021-01-07
**Final Decision**

**Decision:**

Reject

**Comment:**

In this paper, the authors draw connections between probabilistic graphical models (specifically LWF chain graphs) and neural network models. There was general agreement amongst the reviewers that this is an interesting topic that merits further study, and would be of interest to the ICLR audience. At the same time, all of the reviewers have read the author response and there is a consensus that the novelty and significance of this work is limited. The connections between CGs and NNs are somewhat standard and well-known, and the significance of the results has not been convincingly demonstrated.